Sports training enhances visuo-spatial cognition regardless of open-closed typology

Chueh Ting-Yu 1
Huang Chung-Ju 2
Hsieh Shu-Shih 1
Chen Kuan-Fu 1
Chang Yu-Kai 3
Hung Tsung-Min ernesthungkimo@yahoo.com.tw 1
1 Department of Physical Education, National Taiwan Normal University , Taipei , Taiwan
2 Graduate Institute of Sport Pedagogy, University of Taipei , Taipei , Taiwan
3 Graduate Institute of Athletics and Coaching Science, National Taiwan Sport University , Taoyuan , Taiwan
Bertollo Maurizio
Electronic publication date: 2017 May 23
Publication date: 2017
Volume: 5
Electronic Location ID: e3336
Received 2017 Jan 4; Accepted 2017 Apr 19
Copyright: ©2017 Chueh et al.
Copyright year: 2017
Copyright holder: Chueh et al.
License: This is an open access article distributed under the terms of the Creative Commons Attribution License, which permits unrestricted use, distribution, reproduction and adaptation in any medium and for any purpose provided that it is properly attributed. For attribution, the original author(s), title, publication source (PeerJ) and either DOI or URL of the article must be cited.
License URL: https://creativecommons.org/licenses/by/4.0/

Keywords: Cognitive function, Sports, Expertise, Event-related potential

Funding: Aim for Top University Project This research was partly supported by a grant from the “Aim for Top University Project” of the Ministry of Education, Taiwan. There was no additional external funding received for this study. The funders had no role in study design, data collection and analysis, decision to publish, or preparation of the manuscript.

==============================
The aim of this study was to investigate the effects of open and closed sport participation on visuo-spatial attention and memory performance among young adults. Forty-eight young adults—16 open-skill athletes, 16 closed-skill athletes, and 16 non-athletes controls—were recruited for the study. Both behavioral performance and event-related potential (ERP) measurement were assessed when participants performed non-delayed and delayed match-to-sample task that tested visuo-spatial attention and memory processing. Results demonstrated that regardless of training typology, the athlete groups exhibited shorter reaction times in both the visuo-spatial attention and memory conditions than the control group with no existence of speed-accuracy trade-off. Similarly, a larger P3 amplitudes were observed in both athlete groups than in the control group for the visuo-spatial memory condition. These findings suggest that sports training, regardless of typology, are associated with superior visuo-spatial attention and memory performance, and more efficient neural resource allocation in memory processing.

Introduction

A growing body of evidence shows that exercise training positively affects cognitive function (Hillman, Erickson & Kramer, 2008), particularly that which demands greater executive control. Athletes have superior sport performance and physical fitness due to prolonged engagement in sports training. According to the broad transfer hypothesis, extensive practice of specific skills can improve cognition for circumstances outside the specific sport context (Furley & Memmert, 2011). One meta-analytical study has shown that athletes perform better on not only cognitive tasks with sport-related contexts relative to non-athletes, but also general cognitive tasks (Voss et al., 2010). Studies have also found that athletes outperform non-athletes in general executive control paradigms which tap on motor inhibition and cognitive flexibility (Alves et al., 2013; Chan et al., 2011; Taddei et al., 2012; Verburgh et al., 2014).

Schmidt & Wrisberg (2008) suggest that sports can be categorized into two types: open-skill (e.g., racket sports, team sports) and closed-skill (e.g., jogging, swimming, cycling), depending on the variability, predictability, and complexity of the performing environment. Wang et al. (2013) found that open-skill sport athletes (tennis players) demonstrated better response inhibition than closed-skill sport athletes (swimmers). However, Jacobson & Matthaeus (2014) found that closed-skill sports athletes exhibited larger cognitive benefits in interference control tasks than open-skill sports athletes, whereas open-skill sports athletes outperformed closed-skill sports athletes in the problem-solving task. These results suggest that the effect of sports types on cognitive benefit may depend on the specific cognitive domains. Therefore, further examination of the relationships between sports types and different aspects of executive control among athletes is warranted.

Visuo-spatial cognition is one aspect of higher-order cognition worth studying with respect to sports training. Visuo-spatial cognition consists of a multi-faceted set of functions, such as perception, selection, organization, and utilization of location and object-based information, and offers a structure for how we interact with our physical environments (Possin, 2010). Athletes are required to place high demands on visuo-spatial processing when they perform in the field (Yarrow, Brown & Krakauer, 2009). Recent studies have found that sports training, particularly open-skill sports, might enhance cognition with higher visuo-spatial cognitive demand. For example, Wang, Guo & Zhou (2016) found that table tennis athletes exhibited better performance relative to non-athletes when they performed the attentional network test (ANT). Additionally, Wang et al. (2015b) found that badminton athletes not only exhibited shorter reaction times in both visuo-spatial attention and memory conditions than non-athletes, but also showed greater task-elicited modulations in beta power in the attention condition as well as in theta and beta power in the memory condition. These results suggest that open-skill training could facilitate visuo-spatial attention and memory performance at both behavioral and neuro-electrical levels.

Closed-skill sports with a high demand on physical fitness, cardiorespiratory fitness (CRF) in particular (e.g., distance running, swimming, and triathlon), may also benefit visuo-spatial cognition. Several studies with different research designs have shown the benefits of CRF training (e.g., running, cycling) on visuo-spatial attention and memory. For example, one intervention study has shown that six weeks of CRF training (i.e., running) enhanced visuo-spatial memory among young adults (Stroth et al., 2009). A cross-sectional study also showed that young adults with higher CRF demonstrated faster response relative to those with lower CRF counterparts in the visuo-spatial attention task (i.e., Posner paradigm) (Wang et al., 2015a). These results suggest benefits to visuo-spatial attention and memory from both open-skill as well as closed-skill sport training with high demand on CRF. However, none of the previous studies concurrently compared open- and closed-skill training on visuo-spatial attention and memory function.

The employment of electrophysiological measures with high temporal resolution, such as event-related potential (ERP), can provide further insight regarding the effects of sports training on cognition. ERP offers a finer evaluation of distinct cognitive operations occurring between stimulus encoding and response execution (Luck, Woodman & Vogel, 2000). The P3 (b) component is the most frequently examined ERP component. It is the largest positive-going peak waveform occurring approximately around 300 ms following stimulus onset, reflects cognitive processing when attention and memory mechanisms are engaged, and relates to context-updating (Johnson, 1993; Polich, 2007). The amplitude of P3 is associated with the amount of neural resources being allocated to task-relevant stimuli, and latency is related to the timing of stimulus classification (Kutas, McCarthy & Donchin, 1977; Polich & Kok, 1995). Previous ERP studies in athletes focused mainly on response inhibition (i.e., Go-Nogo paradigm) and showed that athletes not only exhibited shorter reaction times than non-athletes, but also larger amplitudes of the N2 and P3 component in a No-go condition (Di Russo et al., 2006). Similar findings were found in middle-aged fencers (Taddei et al., 2012), and disabled athletes (Di Russo et al., 2010). Altogether, these results suggest that open-skill sport training might facilitate several neural correlates in motor inhibition.

However, to our knowledge, there have been no ERP studies that examine the effects of sport training on visuo-spatial cognition, and the comparison between different sport types. Therefore, the aim of this study was to investigate the effect of open and closed sport participation on visuo-spatial attention and memory performance among young open-skill athletes, closed-skill athletes, and non-athlete controls. This study employed the non-delayed and delayed match-to-sample task with both behavioral and ERP measurements. In this task, participants are required to engage visuo-spatial attention and memory processing that can effectively elicit the P3 component (Müller & Knight, 2002). We hypothesized that athlete groups, regardless of sports typologies (i.e., open-skill and closed-skill), would exhibit shorter reaction times than the control group in both visuo-spatial attention and memory condition, and no existence of speed-accuracy trade off with similar results in accuracy-adjusted reaction times and no accuracy difference among groups. Moreover, both athlete groups would show a larger amplitude and shorter latency of P3 than the control group in both conditions.

Method

Participants

Forty-eight participants were recruited from universities in Taipei. They were divided into three groups based on their sport typology, and the three groups were matched in age and gender: open-skill sports group (OS; n = 16, mean age  =20 ± 1.2 years), closed-skill sports group (CS; n = 16, mean age  =21.1 ± 2.3 years), and non-athletes control group (Con; n = 16, mean age  =20.7 ± 1.1 years). Participants in the OS group were badminton (n = 7) or table tennis (n = 9) athletes while those in CS group were swimming (n = 7), triathlon (n = 1), and distance running (n = 8) athletes. The Con group was comprised of those without sport training experience. There was one left-handed participant in each group based on Edinburgh handedness inventory scores (OS  =66.8 ± 46.7, CS  =77 ± 43.4, Con  =60.5 ± 45.4) (Oldfield, 1971). Athletes in both sport groups were competing at national division 1 level and engaged in intensive training for at least six months preceding their participation in the experiment. All participants met the following criteria: (a) they were non-smokers; (b) had normal or corrected-to-normal vision; (c) did not report diagnosed psychiatric or neurological disorders; (d) did not take medication that would influence central nervous system functioning; and (e) were able to perform physical exercise without discomfort or health risks based on an assessment with the Physical Activity Readiness Questionnaire (PAR-Q). All participants were required to sign the written informed consent approved by the Research Ethics Committee of National Taiwan Normal University (201602HM005).

Procedures

Participants were instructed to visit the laboratory for two testing sessions. All sessions were completed within one month and separated by at least one week. Participants were required to refrain from food and drink consumption, except water, 1.5 h before each session. In the first session, participants first completed the demographic questionnaire, socio-economic status of the family (SES) (Hollingshead & Redlich, 1958), handedness inventory (Oldfield, 1971), PAR-Q, International Physical Activity Questionnaire (IPAQ) (Liou et al., 2008), and informed consent form. Then, participants were instructed to sit on a comfortable chair and fitted with an electrode cap in a quiet and dimly lit data acquisition room. Afterwards, participants were provided cognitive task instructions and performed practice trials. The formal data recording commenced when participants reached an accuracy rate of 80% in the practice trials. In the second session, participants were administered a non-verbal IQ test using Raven’s Progressive Matrices: SPM Plus Sets (Styles, Raven & Raven, 1998). Next, participants’ height and weight were measured, and the cardiorespiratory fitness measurement was administered. Participants were given USD $30 compensation after they completed the second session.

Measures

Cardiorespiratory fitness assessment

Cardiorespiratory fitness by peak oxygen consumption (VO2 peak) was measured for each participant utilizing the Bruce Treadmill Protocol, which is a maximal graded exercise test (GXT) on a motorized treadmill. During this protocol, both the speed and slope increased every 3 min until participants were exhausted, and the test was terminated when at least two of following three criteria were met: (a) a plateau in VO2 with increasing exercise intensity; (b) a respiratory exchange ratio above 1.10; and (c) HRmax within 15 beats of age-predicted HRmax (220-age) (American College of Sports Medicine, 2006; Howley, Bassett & Welch, 1995).

Cognitive assessments

This study employed a modified non-delayed and delayed match-to-sample test adapted from Wang & Tsai (2016) to examine visuo-spatial attention (non-delayed condition) and visuo-spatial memory (delayed condition) function (see Fig. 1). The task was programmed with STIM 2.0 software (Neuroscan Ltd, El Paso, TX, USA). All stimuli were presented on a 17-inch computer monitor that was placed 60 cm in front of participants. The stimuli consisted of a red dot (0.5° × 0.5°) randomly presented within a 3.8° × 7.4° gray rectangle. One dot could appear in any one of nine locations (i.e., center, center right, center left, upper center, upper right corner, upper left corner, lower center, lower right corner, and lower left corner) within its rectangle. Participants were instructed to determine whether the location of the red dots appeared in the same position within their respective rectangles.

Figure 1 Illustration of the non-delayed and delayed match-to-sample task.

In the visuo-spatial attention condition (non-delayed), two rectangles were presented simultaneously; one rectangle was placed in the center of the screen, while the other was placed either to the left or to the right of the center. The two rectangles were presented for 180 ms, a duration shorter than is typical for voluntary saccades, to minimize the potential effects of unwanted saccades on the results (Müller & Knight, 2002). In the visuo-spatial memory condition (delayed), the stimulus 1 (S1) was presented for 180 ms with an equal probability on either the left or right of the central fixation (0.5° × 0.5°), followed by a 3-s delay. Stimulus 2 (S2) then appeared for a duration of 500 ms in the center of the screen. Participants were instructed to retain the position of the S1 red dot in their memory during the 3-s delay and then determine whether its position was identical to the position of the red dot in S2.

The response time windows were 2,000 ms for both conditions. Participants pressed the “YES” button with their left thumb when the position of red dots within each gray rectangle were identical and they pressed the “NO” button with their right thumb when they were not. Participants were provided feedback on each response (‘correct’, ‘incorrect’) immediately after the 2000 ms response period. Before the formal test, participants were reminded that accuracy and speed were equally important, and instructed to achieve 80% of response accuracy on the practice trials. A total of 240 trials were equally divided into four blocks, and were randomly presented non-delayed and delayed conditions with equal probability. Rest intervals between blocks were between three to five minutes.

The behavior data were analyzed to derive the response accuracy, median reaction times (RT), and the intra-individual variability in RT, for evaluation of behavior performances. The median RT was measured on correct trials and intra-individual variability in RT using the intra-individual coefficient of variation formula (ICV = the SD of RT/the mean of RT). We used median RT to alleviate potential artifacts deriving from higher rates of outliers that disproportionally contribute to mean RT. In addition, RT distributions are usually positively skewed. Median RT is less sensitive to the skew of distribution (Baayen & Milin, 2010). The response accuracy was calculated as the ratio between number of correct responses and total number of trials. In addition, the accuracy-adjusted RT was computed using the median RT/accuracy rate formula to avoid the potential influence of a speed-accuracy trade-off strategy on task performance (Sutherland & Crewther, 2010).

Electroencephalographic recording

Electroencephalographic (EEG) activity was recorded with 30 electrode sites using an elastic electrode cap (Quick-Cap; Compumedics Neuroscan, Inc., Charlotte, NC, USA). The electrode sites were mounted according to the modified International 10–10 System (Chatrian, Lettich & Nelson, 1985). The electrooculographic (EOG) activity was measured by using four electrodes placed at the outer canthus of each eye, and above and below the left orbit. An average of the mastoid (A1, A2) served as the reference, and FPz was set as the ground electrode on the Quick-Cap. All electrode impedances were below 5 kΩ. The EEG data acquisition was performed with a sampling rate of 1,000 Hz, using a DC- to 200-Hz filter and a 60-Hz notch filter.

For data reduction, the EOG activity was corrected using the algorithm described by (Semlitsch et al., 1986). Epochs were defined as 100 ms pre-stimulus to 1,000 ms post-stimulus, and baseline corrections were performed using the 100-ms pre-stimulus interval. The data were filtered using a 30-Hz low-pass cutoff (12 dB/octave), and with an amplitude outside the range of ±100 µV were excluded at any electrode. After visual inspection, only trials with correct responses were averaged. The P3 amplitude was measured defined as the first maximal positive peak around or after 300 ms following stimulus onset in non-delayed condition and stimulus 2 onset in delayed condition and was measured from the midline electrode sites (i.e., Fz, Cz, Pz) for each participant, and latency was detected as the time point corresponding to the maximum peak amplitude.

Statistical analysis

Data analyses were performed using the SPSS 21.0 software system. One-way ANOVAs were separately computed to test homogeneity of the demographic variables height, weight, non-verbal IQ, handedness scores, socio-economic status (SES) of the family, video game experience, physical activity level, and cardiorespiratory fitness among groups. The independent t-test was conducted to compare the training experience and daily training hours for the past six months between two athlete groups. The two way Group (OS, CS, and Con) × Condition (Non-delayed and Delayed) repeated-measures ANOVAs were separately performed on behavioral data (i.e., median RT, ICV, response accuracy, and accuracy-adjusted RT) to examine group differences in behavioral performance. The three way Group (OS, CS, and Con) × Condition (Non-delayed and Delayed) × Site (Fz, Cz, and Pz) repeated-measures ANOVAs were performed on the P3 amplitude and latency to examine group differences in neuro-electrical performance. Post-hoc comparisons were conducted using LSD significant difference tests. An alpha = .05 was set as the level of statistical significance for all analyses. Eta-squared effect sizes (η2) were reported for significant main effects and interactions, and a Greenhouse-Geisser correction was used to adjust for violations of the sphericity assumption.

Results

Demographic data

Table 1 presents the participants’ characteristics. No significant differences were observed in height (F(2, 45) = 0.158, p = .854), weight (F(2, 45) = 0.522, p = .597), handedness scores (F(2, 45) = 0.536, p = .589), hours playing video games per week in past years (F(2, 45) = 2.256, p = .144), and socio-economic status of the family (F(2, 45) = 2.415, p = .101) among groups. There was a significant difference in non-verbal IQ (F(2, 45) = 11.70, p < .05, η2 = .342), and a post hoc comparison revealed that the control group had a higher non-verbal IQ than both athlete groups, but no significant difference between the two athlete groups was observed.

Table 1 Demographic and physical characteristics of the participants in each group.

Variables	Open-skill (n = 16)	Closed-skill (n = 16)	Control (n = 16)	Total (n = 48)	
Female	7	7	7	21/48	
Left-handed	1	1	1	3/48	
Age (years)	20 (1.2)	21.1 (2.3)	20.7(1.1)	20.6 (1.6)	
Height (cm)	170.2 (9.4)	170.7(6.7)	169.0 (9.1)	170.0 (8.3)	
Weight (kg)	63.9 (11.8)	61.5 (10.1)	59.9 (11.7)	61.8 (11.1)	
Non-verbal IQ test	38.3 (4.2)a	41.7 (5.1)	46.7 (5.4)	42.2 (5.9)	
Socio-economic status of family	2.1 (0.8)	2.5 (0.9)	1.9 (0.7)	2.2 (0.8)	
Video game experience in recently six months (week/hours)	6.3 (3.3)	6.9 (3.5)	10.9 (4.7)	8.5 (4.4)	
Training years	10.8 (2.2)	9.7 (3.2)	0	10.2 (2.8)	
Daily training hours in recently six months	8.7 (1.3)a	12.3 (5.3)	0	10.2 (2.8)	
Total Physical activity level (MET)	9078.6 (2257.1)a	9154.0 (3642.9)	1702.2 (1234.2)	6645.2 (1234.2)	
Cardiorespiratory fitness (ml/kg/min)	46.2 (7.2)a	55.8 (11.9)	39.6 (9.7)	47.2 (11.7)	
Notes.

The number in parentheses is the standard deviation.

a Group effect.

MET, Metabolic equivalent.

With regards to sport characteristics, there was no difference between the two athlete groups in terms of the number of years engaged in sport training (t(30) = 1.094, p = .283). The closed-skill group had longer daily training hours within the past six months than the open-skill group (t(16.659) =  − 2.64, p < .05, η2 = 0.188). Both athlete groups had greater physical activity levels than the control group (F(2, 45) = 47.142, p < .05, η2 = .677), but no significant difference existed between the two athlete groups. Furthermore, there were significant differences in cardiorespiratory fitness across the three groups (F(2, 45) = 11.10, p < .05, η2 = .33), and a post-hoc comparison demonstrated that the closed-skill group had the highest cardiorespiratory fitness, followed by the open-skill group and the control group, but there was only a marginal difference between the open-skill and control group (F(1, 30) = 4.865, p = .06).

Figure 2 Behavioral data of delayed and non-delayed conditions for each group (Mean ± SD).

(A) Main effect of group and condition of the RT. (B) Main effect of condition of the Accuracy Rate. (C) Main effect of group of the Accuracy-adjusted RT. (D) Main effect of condition of the ICV in RT.

Behavioral data

Figure 2 presents the results for response accuracy, RT, ICV, and accuracy-adjusted RT. The response accuracy revealed a Condition effect (F(1, 45) = 93.398, p < .05, η2 = .447), with a higher accuracy in the non-delayed condition (95.91%) than in the delayed condition (89.05%). There were no significant effects of Condition by Group (F(2, 45) = .310, p = .735), and Group (F(2, 45) = 0.025, p = .096) for accuracy results. Furthermore, the RT results revealed the main effects of Condition (F(1, 45) = 9.61, p < .05, η2 = .166) and Group (F(2, 45) = 5.11, p < .05, η2 = .185). RT in the delayed condition (658.76 ms) was shorter than in the non-delayed condition (690.71 ms) and both athlete groups exhibited shorter RT than the control group regardless of the condition, but there was no difference between two athlete groups (OS: 632.52 ms & CS: 655.03 ms < Con: 736.66 ms). There were no significant effects of Condition by Group (F(2, 45) = 1.576, p = .218) for RT results. Regarding ICV, only a significant effect in Condition (F(1, 45) = 50.379, p < .05, η2 = .511) was observed, and the delayed condition (0.242) had a higher ICV than the non-delayed condition (0.198). There were no significant effects of Condition by Group (F(2, 45) = 1.499, p = .234), and Group (F(2, 45) = 0.113, p = .894) for ICV results. Similar results were found for RT and accuracy-adjusted RT, with a significant effect of Group (F(2, 45) = 4.988, p < .05, η2 = .181). Both athlete groups had a shorter accuracy-adjusted RT than the control group regardless of condition, but there was no difference between the two athlete groups (OS: 6.85ms/% & CS: 7.11ms/% < Con: 8ms/%). There were no significant effects of Condition by Group (F(2, 45) = .817, p = .448, and Condition (F(1, 45) = 1.92, p = .173).

An additional analysis was performed on RT that included cardiorespiratory fitness and daily training hours as a covariate to compare the two athlete groups by using two-way Group (OS and CS) × Condition (Non-delay and Delayed) repeated-measures ANCOVAs. The result also showed no significant difference between the two athlete groups (F(1, 28) = 1.485, p = .233) (OS: 622.05 ms & CS: 665.5 ms).

Figure 3 Grand average ERP at Fz, Cz, and Pz sites stratified by group for Delayed condition (A–C) and Non-delayed condition (D–F).

ERP data

Figure 3 illustrates the grand average ERP results at Fz, Cz, and Pz for each group and each condition. For the P3 amplitude, there was a significant main effect of Electrode (F(1.419, 63.875) = 121.113, p < .05, η2 = .725), and an interaction of Condition and Group (F(2, 45) = 3.453, p < .05, η2 = 0.129). A post hoc comparison revealed that Pz (11.67 μV) had the largest amplitude, followed by Cz (7.52 μV) and Fz (3.512 μV). Further decomposition of the Group by Condition interaction revealed a significant Condition effect. Both athlete groups had a larger amplitude than the control group in the delayed condition (F(2, 45) = 4.520, p < .05, η2 = .167) (OS: 8.157 μV & CS: 9.301 μV > Con: 5.244 μV), but there was no significant difference in the non-delayed condition (F(2, 45) = 0.864, p = .428). There was no significant effect of Condition (F(1, 45) = 0.065, p = .801) or interactions of Group and Electrode (F(4, 90) = 0.502, p = .734), Electrode and Condition (F(1.61, 72.441) = 3.070, p = .063), or Group, Electrode and Condition (F(4, 90) = 1.014, p = .405).

Regarding the P3 latency, there were no significant effects of Electrode (F(1.653, 74.371) = 0.802, p = .431) or Condition (F(1, 45) = 0.394, p = .534) or Group (F(2, 45) = 0.033, p = .968) or interactions of Group and Electrode (F(4, 90) = 1.283, p = .282), Condition and Electrode (F(1.542, 69.398) = 2.061, p = .146), Group and Condition (F(2, 45) = 0.018, p = .983), or Group, Condition and Electrode (F(4, 90) = 1.016, p = .404).

An additional analysis was performed on the P3 amplitude that included cardiorespiratory fitness and daily training hours as a covariate in the delayed condition to compare the two athlete groups by using two-way Group (OS and CS) X Sites (Fz, Cz, and Pz) repeated-measures ANCOVAs. The result also showed no significant difference between the two athlete groups (F(1, 28) = 0.009, p = .926) (OS: 8.81 μV & CS: 8.65 μV).

Relationship between RT and the P3 amplitude averaged across midline sites

The correlation analysis was performed to examine whether RT and accuracy-adjusted RT were correlated with the P3 amplitude averaged across midline sites (i.e., Fz, Cz and Pz) in each condition among all participants. In the delayed condition, the P3 amplitude were significantly correlated with RT (r =  − .349, p < .05) and accuracy-adjusted RT (r =  − .313, p < .05), but not in the non-delay condition for RT (r =  − .112, p = .447) and accuracy-adjusted RT (r =  − .126, p = .392).

Discussion

The aim of this study was to investigate the effects of open and closed sport participation on visuospatial attention and memory performance using behavioral and neuro-electrical measures among young open-skill athletes, closed-skill athletes, and non-athletes controls. The main findings were that regardless of their sport typology, athletes exhibited shorter reaction times than the non-athletes in both the visuo-spatial attention and memory conditions, and no existence of speed-accuracy trade off with similar results in accuracy-adjusted reaction times and no accuracy difference among groups. Furthermore, both athlete groups demonstrated a larger P3 amplitude in the visuo-spatial memory condition relative to the control group. Our findings suggest that both open- and closed-skill sport training are associated with superior visuo-spatial attention and memory performance, and better neural resource allocation during memory neurocognitive processing.

Both athlete groups outperformed the non-athletes group on tasks that required visuo-spatial attention and memory processes, which advances existing knowledge. The current study was consistent with previous findings showing that athletes exhibit superiority not only in sports-related contexts (Mann et al., 2007) but also in general cognitive functions (Alves et al., 2013; Jacobson & Matthaeus, 2014; Taddei et al., 2012; Verburgh et al., 2014; Wang et al., 2013). Given that previous studies in athlete and cognition have focused on the executive function domains with less visuo-spatial demands, the present study adds to the literature by employing a cognitive task that requires more engagement of visuo-spatial processing (Wang et al., 2015b). Furthermore, the employment of a cognitive task with conditions that varied in processing complexity verified that athletes committed to prolonged open- and close-skilled sport training are associated with superior visuo-spatial attention and memory processing at both the perceptual and imperative levels.

The finding of a larger P3 amplitude in both athlete groups, compared to the non-athletes group, suggest that athletes invested greater neural resources for the evaluation/classification of imperative stimuli during the retrieval phase of visuo-spatial memory condition than the controls. These results are consistent with previous work focusing on motor inhibition which found that open-skill athletes (i.e., fencers) exhibited a larger P3 component during a No-go condition (Di Russo et al., 2006; Taddei et al., 2012). Within the present study, the P3 amplitude was significantly correlated with RT and accuracy-adjusted RT in the visuo-spatial memory condition, but not in the visuo-spatial attention condition. These findings suggest that sports training might facilitate visuo-spatial memory performances, at least in part, by the modulation of neural resource allocation to task-relevant stimuli. Moreover, the finding of no difference in P3 amplitude between the open- and closed-skill groups implies that prolonged sport training, irrespective of training modality, is associated with enhanced neural resources allocation during the visuo-spatial memory processes.

Notably, the absence of a training effect on the P3 amplitude during the visuo-spatial attention condition is worthy of further exploration. Previous studies have indicated that fencers demonstrated a larger P3 amplitude in No-go condition relative to non-athlete controls, whereas no group difference in P3 amplitude was observed for Go stimuli during a Go/no go paradigm (Di Russo et al., 2006; Taddei et al., 2012). This indicates that the cognitive benefits of training effects might be observed with higher mental loads. In terms of visuo-spatial processing, a greater investment of cognitive resources is required for memory than for attention (Gazzaley & Nobre, 2012). These results, in concert with our findings, indicate that sport training-elicited benefits for neural resource allocation may be specific to cognitive domains with higher mental loads (i.e., visuo-spatial memory).

With respect to P3 latency, the current study found no significant group differences. This result was consistent with that of a previous study which utilized a Go/No-go paradigm (Taddei et al., 2012) and found no significant difference in the P3 latency between athletes and controls, whereas a group difference was revealed in the P3 amplitude. In addition, Wang & Tsai (2016) found that individuals with higher levels of physical activity exhibited a larger P3 amplitude than those with lower levels during visuo-spatial processing, but no group difference was found in the P3 latency. Accordingly, we speculate that sport training enhances visuo-spatial cognitive performances, particularly those with higher mental demand, through the modulation of neural resource allocation, not the speed of stimulus evaluation/classification.

Unlike previous studies (Wang & Tsai, 2016; Wang et al., 2015b) that demonstrated how the memory (delayed) condition had a higher accuracy and a longer RT than the attention (non-delayed) condition across groups in a non-delayed and delayed match-to-sample task, the current study showed that the RT was shorter in the memory condition than the attention condition, which is inconsistent with the results of past studies. We also revealed that the memory condition exhibited higher intra-individual variability in RT (ICV) than the attention condition. As far as the authors are concerned, these results could be interpreted in relation to the higher unpredictability and complexity of the imperative stimulus during the memory paradigm, which resulted in higher uncertainty and impulsive responses, as indexed by greater intra-individual ICV and shorter RT.

There are several limitations of this study. First, its cross-sectional design prevents causal inferences. Second, the current study applied non-delayed and delayed match-to-sample tasks that included visual-spatial attention and memory processing, which omitted the examination of other high-order cognitions. Third, we could not exclude all confounding factors that could bias the relationship between sport type and cognition despite the control of several confounding variables, such as CRF and daily training hours, among two athlete groups. The current study recruited athletes that were mostly from the department of physical education, in which case they may have participated in different types of exercise in addition to formal training. Previous studies have shown that participation in exercise is beneficial to cognitive function (Guiney & Machado, 2013). Accordingly, it is possible that recreational exercise participation played a critical role in biasing the association of sport typology with cognitive function in athletes, which should be carefully considered in future related studies. Fourth, the response sides were not counterbalanced. Although this issue is unlikely to affect our observed effect because all the behavioral measures were pooled across the response side in the present study, future research should consider counterbalancing the response side. Finally, we recruited collegiate athletes, which may limit the generalizability of findings to athletes from different ages (e.g., adolescents).

Conclusions

In conclusion, the current study demonstrated that regardless of sport typology, athletes exhibited superior visuo-spatial attention and memory performance relative to non-athletes at the behavioral level. Furthermore, the training-elicited benefits can be extended to neuro-electrical level of visuo-spatial memory processing. Our findings not only provide convergent evidence for the broad transfer hypothesis that athletes’ expertise can be transferred from sports-specific contexts to general cognitive contexts, but also shed light on the association of sport type with superior visuo-spatial attention and memory performances and greater allocation of neural resource during memory processing.

Supplemental Information

Supplemental Information 1 Type 1=OS, 2=CS, 3=Con

Raw data includes demographics, behaviors, and ERP data for each group, and appears in Table 1, Figure 2, and Figure 3.

Click here for additional data file.

Additional Information and Declarations

Competing Interests

Author Contributions

Human Ethics

Tsung-Min Hung is an Academic Editor for PeerJ.

Ting-Yu Chueh conceived and designed the experiments, performed the experiments, analyzed the data, wrote the paper, prepared figures and/or tables, reviewed drafts of the paper.

Chung-Ju Huang and Tsung-Min Hung conceived and designed the experiments, contributed reagents/materials/analysis tools, reviewed drafts of the paper.

Shu-Shih Hsieh performed the experiments, wrote the paper, reviewed drafts of the paper.

Kuan-Fu Chen performed the experiments, prepared figures and/or tables, reviewed drafts of the paper.

Yu-Kai Chang reviewed drafts of the paper.

The following information was supplied relating to ethical approvals (i.e., approving body and any reference numbers):

This study was approved by the Research Ethics Committee of National Taiwan Normal University (201602HM005).

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
