# Peer review of "Sports training enhances visuo-spatial cognition regardless of open-closed typology"

_PeerJ, doi:10.7717/peerj.3336_

## Round 0.1 · original submission · Major Revisions

We had two very good reviews. The reviewers found your manuscript very interesting and suggested how to improve it. I hope you could address all the requests.

·

Basic reporting

The English language should be improved, as well as the study background and the related references.
The quality of the tables should be improved.
The structure of the manuscript follows the guide to authors.

Please, refer to "General comments to authors" for details.

Experimental design

The manuscript agrees with the Aim and Scope of an original research article of the journal.

The research question is well defined, but the results are not really relevant and able to fill the gap in existing literature. Further, the research lacks to explain why the results could fill this gap.

The investigations are performed following the ethical standards.

The method section need more details.

Please, refer to "General comments to authors" for details.

Validity of the findings

Novelty of the findings not well assessed.

The presented ERP data are not robust.

Conclusions are speculative and not well supported by results.

Additional comments

Title

The title is misleading, because a reader may assume that sport modes are associated with different cortical processing during visual-spatial tasks. This conclusion is not supported by results, wherein the studied ERP component (the P300b) is different in the WM task only. I would strongly suggest changing it according to the real results.

Abstract

Lines 8-10: I would suggest rephrasing this important methodological part of the abstract to better explain both the task and the procedure.
More: “sports-related” should read “sport-related”. The “s” is missed with the “-“.
“, where behavioral” should read “while behavioral”.
“P3 component were recorded”: the EEG is recorded and the P3 is analyzed. Further, P3 is always preceded by the article “the”.
Lines 14-15: The conclusions are a bit speculative. Differences between athletes and controls are obtained at behavioral level (only the RTs) and at neurophysiological level (only larger P3 amplitude during the WM task in athletes than controls). “better fundamental cognition” is far from the nomenclature used in cognitive studies. It does not mean anything. Furthermore, “better fundamental cognition at behavioral and neurophysiological level” is absolutely meaningless. Please, rewrite the entire sentence according to the obtained results and the relative cognitive interpretation.

Introduction
Overall, the introduction needs to be reviewed. In particular, the first paragraph introduces the study mentioning data from children and older adults; the second paragraph presents literature on athletes (this is well-done); the third paragraph refers to the cognitive benefits induced by open-skill sports when compared to CLOSED-skill (not close-skilled) sports; the fourth paragraph shifts its attention to the benefits of cardiovascular fitness on visual-spatial cognition; from the fifth on, we discover the study’s aims. There is a gap within the first part of the introduction (paragraph 1 to 4), where the literature is reported without a link to the study. Further, the authors should focus on one part of the available literature. For example, there are numerous studies on the benefit of cardiovascular fitness on cognition in different populations and they reported only a very little number of studies. I would just focus on sport studies. On the other side, the literature on athletes is less mentioned and need to be improved. Finally, the sport ERP literature is lacking of recent and relevant studies (for example see Bianco et al., 2017). More details are provided below.
Page 2
Line 2: executive function: the executive function is not ONE function or process, but it comprises several processes, like inhibition, WM, flexibility, involved in goral-directed behaviors. Please, talk about cognitive processing and executive functions.
Lines 5-10: The first paragraph of the introduction should report the literature particularly relevant for the study. At this point I would not make reference to children and older adults, but rather to young athletes. Please, remove or move later this sentence. Instead, the reasoning beginning at line 11 is much more appropriate here.
Line 16: “interactive athletes”: The adjective does not have the willed meaning. Please change.
Line 22: “interceptive sports”: may be “interceptive actions” not sports. Please, change according to your meaning.
Lines: 28-30: Even not animal studies.
Lines 37-30: Please provide a clear reference on what you mean for visuo-spatial cognition.
Lines 40-41: please rephrase. Also, CRF stays for cardiorespiratory fitness NOT cardiovascular fitness. Please, change accordingly throughout the manuscript.
Page 3
Lines 2-4: please reorganize this sentence, because it is difficult to follow.
Lines 1-7 you refer to the fitness-related benefits in cognitive visuo-spatial processing. Then (lines 8-10) you talk about the benefits of sport on cognition regardless of the CRF. Later (lines 11-14) you summarize the above-mentioned data in a very confusing way. Please, modify this paragraph.
Lines 15-16: “measures of neurophysiological correlates”. Please, rewrite.
Line 18 “stimulus encoding or response preparation”. It is not or, but both processing are evaluated by ERPs. Please change “or” with “and”.
Lines 19-24: Are you sure that the P3 component reflects ONLY those processing? Please, consider also these papers: Leuthold & Sommer, 1998; Verleger, 1997, 2010; Ouyang et al., 2011, 2013, 2015; Saville et al., 2011, 2014, 2015; O’Connell et al., 2012; Kelly & O’Connell, 2013; Twomay et al., 2015; Saville et al., 2012; Nieuwenhuis et al., 2005; Berchicci et al., 2016.
Line 25: “PA”. Please define physical activity somewhere.
Lines 32-38. I would try to report only what you did in the study. For example, “ERP measures”… you investigated only one component. What do you mean for better visuo-spatial cognition? And what for neuro-electrical performance? Superior behavioral? Please, be specific!

Methods
Participants
Lines 3-8: Rewrite like follows: “Participants were assigned to one of the following groups: open-skill sport () if they played…; closed-skill sport…
Line 13 replace “without” with “not”.
Did you explain the study before getting the signed informed consent? I am sure, thus, include this information.

Cognitive assessment
The paradigm needs to be better explained. The authors may first describe the common characteristics of the tasks and than the difference in the delayed.
Lines 5-8: “The stimuli consisted of a red dot”. One red dot, more red dot? And the fixation point?
Lines 28-29: “240 trials, each consisting of 4 blocks of 60 trials”. There is something wrong with this.
Line 34: “all trials” here refers only to response accuracy, whereas it should be referred to all of the behave measures.How do you calculate the response accuracy? And the RT (mean or median? Why?).
EEG recording
Line 41: the EEG cap is “monted” according to the 10-20 IS.
Line 2: did you use FpZ site as ground, since it is placed on the mid-forehead?
Lines10-13: Are you sure that the P3 can be reliably recorded on Fz within this task? What about the latency (you discussed statistics and results without showing how to calculate it)? Why you did not analyze the early visual components (P1 and N1) on parietal occipital sites, given your interest in visual-spatial attention? And what about the CNV in the WM paradigm? I feel like something is missing in this study. Why you did not correlate P3 amplitude/latency with RTs?

Statistical analysis
Line 15: “One way ANOVAs were” instead of “a one-way ANOVA was”.
Here the t-test explanation is missed.

Results
There is inconsistency between demographical data reported in the manuscript and the table, mainly for the reported significances. Please check and revise.
Partial eta squared (as you stated in the statistical part) should be reported for significant effects only.
Table 2 is difficult to understand. Please, change as a post-hoc table.
Figure 2 reports waveforms without specifying the electrode.
Based on ERP results, athletes showed larger amplitude than controls in the WM task only. All of the other effects (such as the site effects) are not relevant to the present study and for any neurocognitive explanation.

Discussion
I think that the discussion should avoid circular arguments. However, I believe that rewriting the introduction would benefit also the way in which the discussion is presented. Again, I would suggest focusing on the literature closely relevant for the interpretation of the real results. Further, since the current phrasing makes comprehension difficult, I would be happy to deeply revise the discussion afterwards.
Lines 4-5: “faster processing speed” why? You should have effect on the P3 latency.

Reviewer 2 ·

Basic reporting

A very well-written and professionally presented manuscript.
One very minor point – in the discussion a greater P3 amplitude is clearly interpreted to reflect the investment of “greater neural resources” (P10 L27). However, it is not so clear in the introduction. Consider revising your sentence on P3 L19-21 to more clearly state that you are interpreting a bigger amplitude to reflect more resources allocated, and a shorter latency to reflect quicker processing. This directional interpretation is not immediately obvious from the current wording.
This could be especially useful since there are other interpretations of P3 in the literature (e.g., predictability, threat, load etc). Hence, being clear would inform readers what you believe the P3 to reflect, for the purpose of this manuscript, from the outset.

Experimental design

A generally sound and well-designed experiment. My only recommendation here is to add a sentence to describe how P3 latency was scored. You describe the amplitude measure on P6 L12-13, but not latency. This minor detail is required to permit replication.

Validity of the findings

No comments.

Additional comments

P10 L10. Have you considered using more precise terminology than “faster processing speed”? For example, assuming latency is interpreted to reflect time required for stimulus detection, there were no differences here. Thus, could this indicate that the athletes’ response time advantage comes specifically from superior response selection and response programming (which seem more precise terms than a generic “faster processing speed”)? You hint towards this yourselves on P11.
***Note that this is a general interest question rather than a firm recommendation, you may have a good reason for choosing the “faster processing speed” terminology. However, you might consider making some minor tweaks if (and only if) you agree with my interpretation and find this comment helpful.

Discussion. I couldn’t find any mention of the P3 latency main effect of electrode that you report on P8 L31. I appreciate this finding wasn’t hypothesised, but seeing as it is reported, readers would probably expect to see it acknowledged and at least a sentence or two to try and reconcile in the discussion. Consider adding brief comment on this topic.

---

## Round 0.2 · Minor Revisions

I regret I am still unable to accept the manuscript in its present form. As you can see below the third reviewer raised many criticism about this 2nd version that I cannot ignore. The first reviewer, even if accepting the manuscript, suggested checking grammar and typos (e.g. line 108 “modalities” and not “motilities”). To be honest also the second reviewer wrote me privately that for many reasons the first version was better.

At this stage, what I suggest is trying to make fluent this version trying to address the concern of the third reviewer. I will check myself the final version without sending it again to the reviewers. I hope to have it back as soon as possible.

In addition to the general review of the manuscript, below you can find some specific suggestion that I hope could help your revision.

1) Title: You should find a more clear title such as Sports training enhances visuo-spatial cognition regardless of open-closed typology (or skill categorization). An alternative could be “Open and closed skill sports training equally enhance visuo-spatial cognition.”
2) Abstract: I suggest changing as proposed “The aim of this study was to investigate the effect of open and closed sport participation on visuospatial Cognition…” AND PLEASE REVISE ACCORDINGLY ACROSS THE MANUSCRIPT
3) Answering to the concern of the third reviewer about the "visual-spatial cognition" concept, I suggest adding that your study focused on visuo-spatial attention and working memory.
4) About hypothesis, please provide more specific statements related to cognitive and cortical mechanisms you expect to highlight with your study, and specifically with VS attention and WM
5) Please check typos and grammar mistakes carefully
6) The manuscript needs an accurate proofreading for consistency and completeness of the information in results section with respect the protocol and the measures provided.

·

Basic reporting

I would thank the authors for the accurate and minute revision of the manuscript. All the points raised by the reviewer have been addressed.
The English has been checked out, the references appropriately referred to, and the results presented in line with the methods and hypothesis of the work.

Experimental design

Experimental design, methods and ethical standards are correctely presented in the revised version of the manuscript.

Validity of the findings

The conclusions of the work are well stated and justified baby the actual results obtained in the study.

Additional comments

I would only suggest to read again the paper to double check the few grammar and language imperfections.

Reviewer 2 ·

Basic reporting

No comment

Experimental design

No comment

Validity of the findings

No comment

Additional comments

The authors have done a nice job at responding to all of the relatively minor comments raised in my previous review. I have nothing further to add.

Reviewer 3 ·

Basic reporting

A major issue with the revised version of the ms is that the text is partly vague. At times it is imprecise or difficult to follow (which may be due to complicated wording). For example, the title has been accomodated but plain speaking of "modes" does not make the content accessible. (The revised version lost pagination which makes it awkward to refer to specific passages.) The introduction in particular is difficult to follow as it lacks coherence to some extend. For example, the finding of table tennis players showing improved visual attention (compared with novices) is put next to the statement that badminton players show enhanced modulations of neural oscillations. I cannot follow such relations and oftentimes asked myself what the authors want to say (for specific phrases).

vague terminology: What is meant by "visual attention executive network tasks" or "attention-related neurocognitive performance?"

The central concept "visual-spatial cognition" is too broad. The authors would need to be more precise in what the are interested in and how it is operationalised.

Some sentences appear to hard to parse. "The order of equal numbers of [...] were randomly presented within experiment."

Most critically, I could not find hypotheses. Where the authors mention hypothesising (very last sentence of the intro), it is actually predictions of the variables of interest. Note that even the predictions do not fit with the analyses (no prediction for some behavioural variable such as accuracy, neither for associations). The control group is only mentioned in the last word of the introduction.

ERP peak detection is described under Recordings, not under Statistical analysis.

Experimental design

The design is reasonable. Handedness scores should have been reported and checked whether they differ among groups (as Edinburgh handedness inventory has been used). However, the selection of participants did not result in a balanced sample which undermines the validity of the results. Less critical but the response side has not been counterbalanced as is standard in such research.

Validity of the findings

For some results, important details/data is missing. At the end of the results correlations have been calculated and one significant correlation has been reported. However, it is not stated for which group the coefficient was significant and for which electrode.

The fundamental issue as, as stated above, that the sample is not well matched. The groups differ in varying patterns for a number of variables which could alternatively be responsible for the RT effects and/or the ERP effects. Unfortunately, this renders the study inconclusive. While "confounding factors" are mentioned in passing, nothing of such limitations is mentioned in the abstract or the conclusion.

For the P3 amplitude and latency, the peak was detected between 300 and 600 ms (time-locked to which stimulus for the delayed task?). Fig. 3 suggests that for the group 'open skill ' the P3 peaks around or after 600 ms (CZ, PZ). Certainly, individual ERPs will vary even more, especially as P3 can vary considerably. So, the ERP plots appear to not fit the description and maybe the peak detection procedure prevented a fair evaluation.

---

## Round 0.3 · accepted · Accept

I believe that your efforts to manage feedback accompanying my previous decision have noticeably improved the manuscript. You and your coauthors have my congratulations.